# Cluster-Flow Parallel Coordinates:
# Tracing Clusters Across Subspaces

Nils Rodrigues* (iD)    Christoph Schulz* (iD)    Antoine Lhuillier†    Daniel Weiskopf* (iD)

Visualization Research Center (VISUS)
University of Stuttgart, Germany

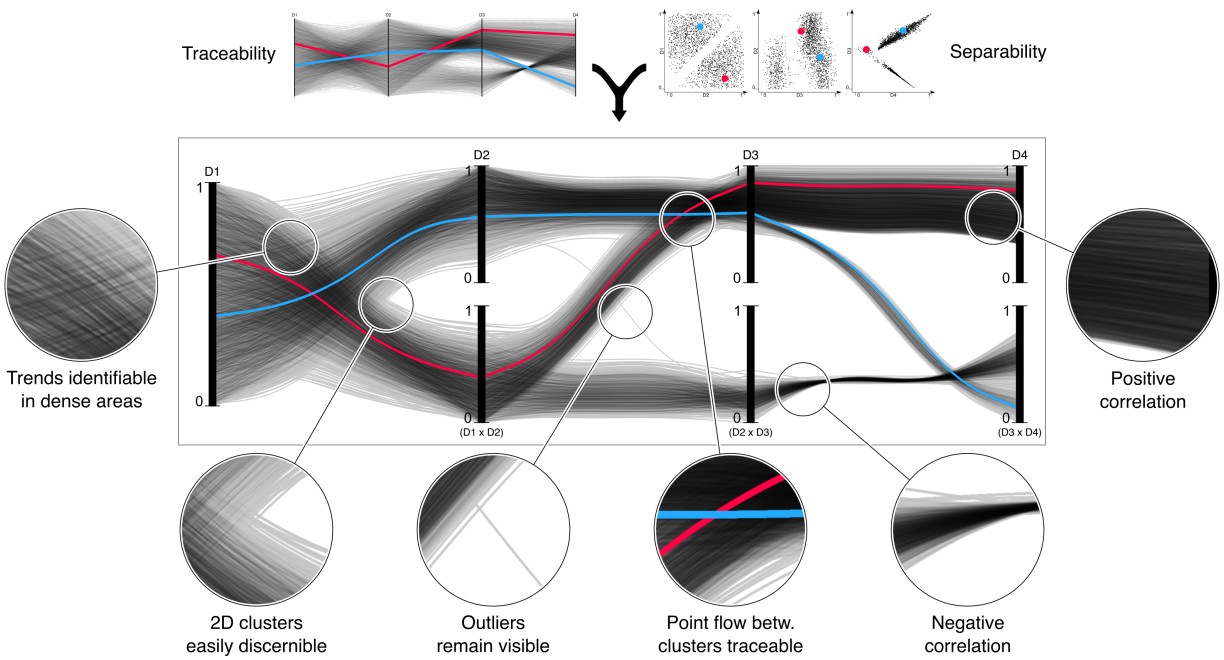

Figure 1: Our Cluster-Flow Parallel Coordinates Plot (CF-PCP) combines advantages of regular Parallel Coordinate Plots (PCPs) and Scatter Plots (SPs) as shown here using 2,000 generated points in dimensions D1–D4. CF-PCPs are read from left to right: The data is grouped by pairwise dimensional clustering, i.e., stacked axes beneath $D_i$ show all clusters from subspace $D_{i-1} \times D_i$. CF-PCPs allow for salient illustration of clusters and traceability across multiple dimensions alike. Thus, we argue that our technique can reveal patters that are difficult to perceive from a linked combination of SPs and traditional PCPs (cf. red and blue data points).

## ABSTRACT

We present a novel variant of parallel coordinates plots (PCPs) in which we show clusters in 2D subspaces of multivariate data and emphasize flow between them. We achieve this by duplicating and stacking individual axes vertically. On a high level, our cluster-flow layout shows how data points move from one cluster to another in different subspaces. We achieve cluster-based bundling and limit plot growth through the reduction of available vertical space for each duplicated axis. Although we introduce space between clusters, we preserve the readability of intra-cluster correlations by starting and ending with the original slopes from regular PCPs and drawing Hermite spline segments in between. Moreover, our rendering technique enables the visualization of small and large data sets alike. Cluster-flow PCPs can even propagate the uncertainty inherent to fuzzy clustering through the layout and rendering stages of our pipeline. Our layout algorithm is based on A*. It achieves an optimal result with regard to a novel set of cost functions that allow us to arrange axes horizontally (dimension ordering) and vertically (cluster ordering).

**Index Terms:** Human-centered computing—Visualization—

---

*e-mail: firstname.lastname@visus.uni-stuttgart.de
†e-mail: antoine.lhuillier@gmail.com

Visualization techniques; Human-centered computing—Visualization—Visualization application domains—Information visualization

## 1 INTRODUCTION

The analysis of multivariate or multidimensional data is a long-standing research topic in visualization [51]. Nowadays, with multivariate data being ubiquitous, a good interplay of automated data analysis and visualization is very important for gaining insights. In the realm of data analysis, subspace clustering allows analysts to find cross-dimensional relationships between data points, leading to useful classification methods [9, 16]. On the other side of the spectrum stands an important class of visualization techniques: parallel coordinates plots (PCPs) [22,27]. They play an important role in visualizing multivariate data as their core concept of parallel axes is easy to grasp and, unlike scatter plots, they scale for increasing dimensionality. Typically, they render each data point as a polyline, curve, or density field [21]. Existing techniques combine PCPs with either a single global cluster assignment for each data point [2] or with clusters in 1D data dimensions [39]. They then resort to edge bundling or color coding to show cluster memberships [22].

In this work, we aim to combine subspace clustering with the visualization advantages of PCPs. To this end, we propose a novel approach that facilitates the visualization of clusters in 2D subspa-

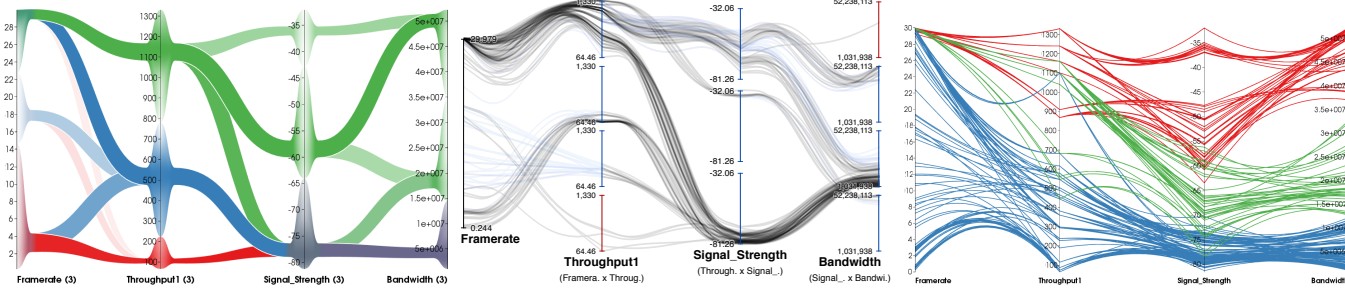

Figure 2: Comparison of different clustering and PCP techniques using the NetPerf data set [49]. The edge-bundling layout (a) replaces lines entirely and only draws a single band between each pair of neighboring 1D clusters [39]. Our cluster-flow parallel coordinates (b) draw individual lines between 2D clusters of neighboring axes. Clusters are arranged vertically with our crossing minimization algorithm from Section 4.3. Illustrative parallel coordinates (c) emphasize clusters over all dimensions using colored lines and force-based bundling [36]. Figures (a) and (c) © 2014 IEEE. Reprinted, with permission, from [39].

ces while still maintaining information about the characteristics of individual data elements in PCPs: the Cluster-Flow Parallel Coordinates Plot (CF-PCP). As depicted in Figure 1, our approach works on two visual levels inherent to the image. On the overview level, a coarse visualization allows users to trace and follow the evolution of subspace clusters across dimensions by duplicating axes for each cluster and stacking them vertically. On the detail level, it maintains the readability of correlation of data points and other data characteristics by ensuring that the incoming and outgoing links reflect the original information from regular PCPs, i.e., our approach keeps the original slopes of each data element. We demonstrate that CF-PCPs allow users to trace both hard and fuzzy subspace clusters across dimensions. Moreover, we propose new metrics to optimize dimension ordering in CF-PCPs based on subspace clusters and to reduce crossings between clusters. Our main contributions are

- a new PCP with **density rendering** for **subspace clustering** that preserves the **readability of correlations**,
- an approach to **visualizing uncertainty** from fuzzy clustering,
- an A*-based algorithm for the **optimal layout** with respect to a novel **set of metrics** that reflect the compatibilities of clusters between dimensions, and
- a sample implementation of CF-PCP[1] using fuzzy DB-SCAN[2] for subspace clustering.

## 2 RELATED WORK

Multivariate or multidimensional visualization is a major and vibrant research area of the visualization community. Respective survey papers are available from Wong and Bergeron [51] and Liu et al. [35]. Our work addresses the visual mapping of multivariate data to PCPs; therefore, the discussion of related work focuses on parallel coordinates, in particular, in combination with clustering. Parallel coordinates for data analysis go back to seminal work by Inselberg [26, 28] and later by Wegman [50]. For a comprehensive presentation of PCPs, we refer to Inselberg's book [27]. Despite its popularity, the underlying geometry of a PCP coupled with a high number of data points can quickly lead to overdraw and thus visual clutter [22]. This makes it hard for users to explore and analyze patterns in the data set. To address these challenges, researchers have investigated cluster visualization and the saliency of underlying patterns.

A first approach to cluster visualization in PCPs is to explicitly compute clusters in the data set and display them using different visual encodings. Inselberg [26] suggested drawing the envelope of the respective lines in parallel coordinates using the convex hull. Fua et al. [14] investigated rendering clusters with convex quadrilaterals resembling the axis-aligned bounding box of a cluster. More recently, Palmas et al. [39] proposed pre-computing 1D clusters for each dimension using a kernel density estimation approach and then linking neighboring axes using compact tubes in which the width encodes the number of data points in the cluster (see Figure 2a). They then used color coding to mark the clusters of a chosen dimension. Their technique produces a highly summarized and largely clutter-free visualization, reminiscent of a stream visualization such as baobab trees [46] as well as timelines by Vehlow et al. [47]. Although visually similar, these techniques differ from our approach as they do not keep the correlation details of data elements encoded in the PCP links, nor do they show the cluster flow between subspaces.

Another approach changes the visual mapping of the lines to implicitly show clusters. Here, edge bundling [33] has shown to be an effective technique that helps users find clusters and patterns within PCPs [20]. Illustrative parallel coordinates [36] bundle PCPs in image space by pre-clustering the data set first (via *k*-means) and then render the lines as curves using B-splines (see Figure 2c). Zhou et al. [54] used a variant of force-directed edge bundling [23] to directly compute the clusters based on the patterns emerging from the bundling algorithm. Heinrich et al. [20] extended previous work [36, 54] by providing $C^1$-continuity between B-splines to emphasize end-to-end tracing. Our approach also changes the visual mapping of the links and reduces clutter. However, our overall composition of links differs because we duplicate axes and focus on keeping information about the correlation of data elements in the clusters.

Another approach to improve PCPs is to change the order of dimensions. While ordering can also be applied to other visualization techniques, it is especially inherent to the construction of PCPs, where the order of axes directly affects the revealed patterns [50]. Pargnostics [8] uses metrics such as the number of crossings, the angle of crossings, or the parallelism between dimensions for ordering. Tatu et al. [45] presented a method to rank axes in parallel coordinates using features of the Hough transform. Ankers et al. [1] proposed a pairwise dimension similarity based on Euclidean distance. Peng et al. [41] defined a clutter-based measure and used it in an A* algorithm to order the dimensions. Ferdosi and Roerdink [12] generalized the notion of dimensional ordering of axes by expanding the concept of a pairwise similarity measure to sub-

[1] https://github.com/NilsRodrigues/clusterflow-pcp
[2] https://github.com/schulzch/fuzzy_dbscan

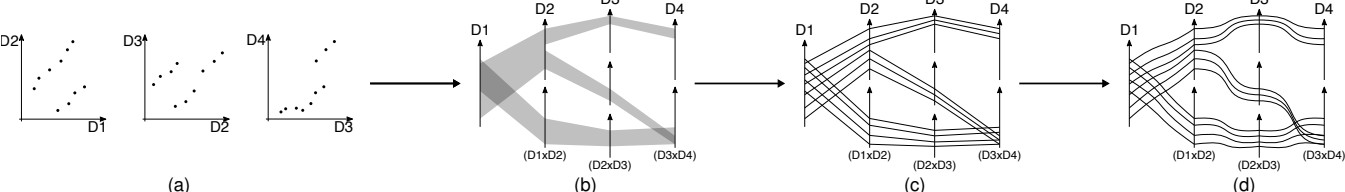

Figure 3: Construction of CF-PCPs. Scatter plots are well suited to show clusters of multivariate data in 2D subspaces (a) but do not readily show cluster relations across more dimensions. We extend PCPs by creating an individually duplicated axis for each cluster to show high-level flow between clusters (b). Showing the underlying data points as individual lines increases the available level of detail (c). To maintain visual data patterns and the perception of correlations, we restore the original PCP's line angles near the axes (d).

space similarity measure using a predefined quality criterion. Later, Zhao and Kaufman [52] proposed several clustering techniques to optimize the ordering of axes in PCPs, e.g., a *k*-means or a spectral approach. Tatu et al. [45] also proposed subspace similarity measures based on dimension overlap and data topology. In general, all these methods focus on subspaces. Our approach differs from the previous ones as our duplication of axes emphasizes flow between clusters and imposes the definition of new similarity measures to order both data dimensions and clusters.

Our paper also addresses the problem of visualizing uncertainty [5, 6, 40] from fuzzy clustering. Techniques for hatching, sketchiness, as a specialized form of spatial uncertainty, have gained a lot of attention [3, 15, 34]. However, there are only few works on uncertainty in the context of parallel coordinates. Dasgupta et al. [7] conceptualized a taxonomy of different types of visual uncertainty inherent to PCPs. Feng et al. [11] focused on showing uncertainty in the data by mapping confidence to saliency in order to reduce misinterpretation of data, relying on density representation of uncertainty. We adopt color mapping to visualize fuzziness of clustering in PCPs.

To the best of our knowledge, there is no previous work that combines all mentioned topics. Kosara et al. [32] also visualize flow but for categorical data instead of clusters. Their portrait layout of PCPs also does not calculate an optimal order for dimensions or categories. Nested PCPs by Wang et al. [48] look very similar but have a global clustering instead of using subspaces. They are meant for ensemble visualization in conjunction with other plots and use global color mapping that only depends on a single dimension.

## 3 MODEL AND OVERVIEW

We first provide an overview and outline intermediate steps of our technique (see Figure 3). We assume that we have a set of data points in a multivariate data set as input: $P = \{\vec{p}_i \in \mathbb{R}^n\}$. An algorithm assigns each of these points a degree of membership to clusters, resulting in tuples $(\vec{p}_i, m_{k,i})$. Here, $m_{k,i} \in [0, 1]$ describes the degree to which data point $\vec{p}_i$ belongs to the cluster with index $k$. Hard clustering is a special case of this, with $m_{k,i} \in \{0, 1\}$. In contrast, soft labeling allows for partial memberships. A single cluster is then defined as $c_k = \{(\vec{p}_i, m_{k,i}) \mid m_{k,i} > 0\}$ and $C = \{c_k\}$ comprises all clusters. We focus on subspace clustering because distance measures lose expressiveness when dimensionality increases, leading to superfluously fuzzy or inconceivable clusters. Hence, we compute clusters for each pair of dimensions $(\mathbb{R}_i, \mathbb{R}_j)$ as shown in Figure 3a, i.e., we look at sets of 2D subspaces. Our current implementation employs FuzzyDBSCAN [25], which is an extension of the classic and popular DBSCAN algorithm, for fuzzy clustering. However, our visualization is independent of the chosen clustering algorithm, it just assumes that clustering provides labels for each data point. The data that serves as source for clustering is shown beneath the axes.

The goal of CF-PCPs is to show the information about subspace clusters and the actual data points alike. On a coarse level, they

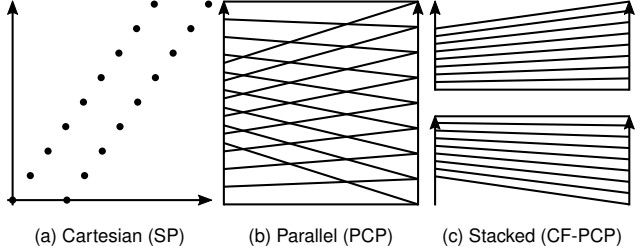

(a) Cartesian (SP)    (b) Parallel (PCP)    (c) Stacked (CF-PCP)

Figure 4: Illustration of two clusters, using different coordinate systems. The scatter plot (a) uses Cartesian coordinates to clearly show them as dots placed along two straight lines. The clusters are hard to distinguish in a regular PCP (b) without additional visual variables, e.g., color. Our cluster-flow layout duplicates PCP axes for each cluster to reduce clutter and increase readability (c).

show the number of clusters in the subspaces and how data elements virtually flow from one cluster to another (see Figure 3b). Our approach to showing this flow is based on axis duplication: instead of a single (vertical) axis for a data dimension, we place clones of this axis on top of each other, i.e., at the same horizontal position. Each data cluster gets its own axis clone. Through this duplication, we can render the stream of data elements between clusters, which are visible even if one does not focus on individual lines in the CF-PCPs but just looks at the visualization as a whole. Section 4 describes the details of the axis duplication and how we can achieve an optimal layout of the data dimensions and cluster ordering.

By rendering an individual line for each data point, we include further details (see Figure 3c) in a fine-grained level of our proposed visualization. We use a curve model (see Figure 3d), as described in Section 5, that allows us to infer correlations and other data characteristics for individual data elements, similar to regular PCPs. Furthermore, we present a density rendering and color mapping approach that allows us to visualize large data sets and uncertainty from fuzzy clustering (see Sections 5.2 and 5.3).

## 4 LAYOUT

The key aspect of our layout of CF-PCPs is the duplication of axes (Section 4.1) because this serves as a basis to visually separate clusters and show the flow between clusters in the different subspaces. Each ordering of axes results in a unique flow pattern between clusters, therefore we present a new method to optimize the horizontal axis order according to patterns in data flow (Section 4.2). Section 4.3 introduces a technique to order the clusters on each data axes (*y*-axis) vertically.

### 4.1 Axis Duplication

CF-PCPs extend traditional PCPs by using the vertical image space to show cluster assignments. Unlike any other PCP variant, we visualize each cluster on its own (local) axis. As Figure 4 shows,

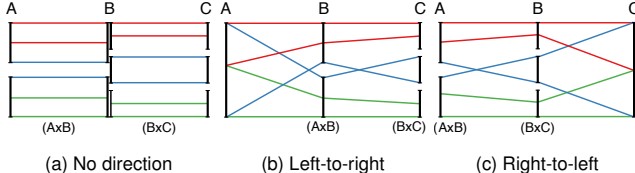

(a) No direction      (b) Left-to-right      (c) Right-to-left

Figure 5: Various reading directions with data points colored according to clusters in $B \times C$. Without a reading direction, we obtain overlapping axes and discontinuous lines (a) because the 2D subspaces $A \times B$ and $B \times C$ do not have the same number of clusters. With a reading direction, axes show the clusters within the subspace of the current dimension and its neighbor to the left (b) or right (c).

we create two duplicates of the axes to draw the two clusters and stack them vertically.

Axis duplication becomes necessary because we aim to show clusters in 2D subspaces, not just in 1D [39]. For the latter case of 1D clustering, we could just squeeze the data vertically to bundle the line into clusters on each axis individually. However, this approach will only work if the clusters are already separable in a single dimension. Figure 4a shows an example with two clusters that are visible in a scatter plot and linearly separable in two dimensions but not along 1D axes. Here, traditional PCPs lead to intertwined visualizations of the two clusters, and even axis scaling could not separate them Figure 4b. With axis duplication Figure 4c, we now see two clearly distinguishable bundles of lines that correspond to the two clusters, i.e., clusters are easy to recognize. However, there is yet another important benefit: the PCP lines can now be inspected for each cluster separately and, thus, we can investigate correlations or other data characteristics independently within each cluster.

Up to now, we only handled a pair of two data dimensions, but parallel coordinates support multivariate data. Figure 5 shows an example sequence with three dimensions. In subspace $A \times B$, there are two clusters, so we create two copies. $B \times C$ contains three clusters, so we draw three copies. This results in overplotted axes for dimension $B$ and interrupted polylines for the data points (see Figure 5a). Therefore, we introduce a reading direction left-to-right (LTR) to create as many copies of axis $B$ as there are clusters in $A \times B$. We duplicate axis $C$ as often as there are clusters in $B \times C$ (see Figure 5b). Since axis $A$ has no pair to the left, there is no explicit clustering and no duplication. While the opposite reading direction works accordingly (see Figure 5c), we chose a consistent LTR layout for all figures in this paper. Beneath the axes, CF-PCPs also show the two data dimensions that were used for clustering to help the viewer identify the reading direction.

Cloning the axes and stacking them vertically increases the plot area and vertically shears the lines that represent the data points, especially for large numbers of clusters. We address both issues by scaling down cloned axes with a root function. Assuming the height $h_1$ of a single regular PCP axis and $n$ as the number of stacked axes, we get a total height of

$$h(n) = n \cdot h_1 \cdot (1/n)^r \, , \qquad (1)$$

where $r \in [0, 1]$ controls the root scaling. Selecting $r = 0$ will result in no downscaling at all, while $r = 1$ will keep the original height without any growth. Spaces between axes are determined with

$$s(n) = s_2 \cdot h_1 \cdot (1/(n-1))^r \, , \qquad (2)$$

where $s_2$ is the percentage of $h_1$ we want to use as space between two duplicates. The remaining space is then used for the actual axis clones. We chose $r = 0.8$ and $s_2 = 0.1$ for figures in this paper to balance between growing height and readability.

Axes in regular PCPs are normalized to only display the used range of each data dimension. Multiple labeled tick marks enable viewers to read values from data points at their intersections

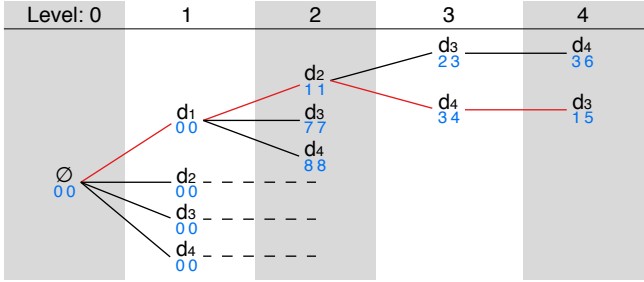

Figure 6: Tree representation of our model for ordering of dimensions $d_1$ to $d_4$. The blue numbers beneath the dimension names represent costs (left: individual; right: accumulated). Levels 0 and 1 have no costs because there are no pairs of clusters yet. The path $d_1$, $d_2$, $d_4$, $d_3$ is optimal as it is at the deepest level and others are at least as expensive. Missing paths at level 3 and 4 have not been expanded yet, because their costs were never minimal.

with the axes. As the number of clusters rises, axes in CF-PCPs shrink. Retaining the same amount of labeled tick marks as in regular PCPs would add more overdraw to the already densely packed area around the axes. Therefore, we limit the ticks marks to the maximum and minimum of the data range in each data dimension and progressively scale the label size according to the space between axes.

### 4.2 Dimension Ordering

Just as with regular parallel coordinates, the visual patterns in a CF-PCP depend heavily on the order of data axes. For this reason, we propose a method for optimizing the order automatically. Previous work investigated various metrics—like number of crossings, angles between lines, correlation strength, and more—to optimize the order [8, 30]. At first glance, we could have simply reused existing algorithms or defined dimension ordering as an overlap-removal problem for the polylines in the PCP. However, CF-PCPs do not only show the underlying data points but also depict flow between subspace clusters on a coarse scale. For example, to calculate the costs of chaining two sets of clusters between display axes $a_i$ and $a_{i+1}$, we also need to know the previous axis $a_{i-1}$. This is due to our 2D subspace clustering and reading direction LTR: clusters shown at $a_i$ are calculated with data from dimensions $d_{i-1} \times d_i$. Only then can we calculate costs with the next axis $a_{i+1}$, which contains clusters from $d_i \times d_{i+1}$.

As shown in Figure 6, we choose a tree $T$ as primary data structure for our proposed order optimization. We start by creating the root node at level 0 and add a child for each data dimension at level 1. Then, we expand each leaf recursively by attaching a node for each unused data dimension. Each level corresponds to an index in the sequence of display dimensions. This way, for $d$ data dimensions we obtain a tree with $d + 1$ levels, where each node has $d - level$ children. The tree contains $d!$ different paths—corresponding to all permutations of the data dimensions. Costs from one set of clusters to the next depend only on three consecutive data dimensions. Thus, there are only $d \cdot (d - 1) \cdot (d - 2)$ different costs to compute, which can be cached and reused in the tree traversal.

We apply the A* algorithm [17] to compute the shortest path in $T$ and implement it by lazily expanding a tree node when it is the leaf with the lowest cumulative costs. The heuristic $d - level$ estimates the remaining costs. This has implications on the metric we use to define the distance between two sets of clusters: The minimum cost from a parent to a child has to be 1 for the heuristic to work. With this model, the A* algorithm is guaranteed to find an optimal solution. We combine lazy evaluation with caching of cost values to compute the A* optimization efficiently.

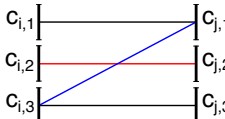

Figure 7: Simplified CF-PCP with cluster sets $c_i$ and $c_j$. Top and bottom (black) never cross any other bundles. A red reference bundle only crosses lines that start below and end above (blue).

Having decided on a basic optimization algorithm, we choose a measure for the costs of displaying clusters next to each other. CF-PCPs are targeted at showing how clusters evolve from one subspace to the other, how the data points flow from one to another. Therefore, we propose a metric that focuses on the similarities and differences of clusters. Let $c_i$ and $c_j$ be two sets of clusters. The number of elements in them are $n = |c_i|$ and $m = |c_j|$. In a first step, we generate a similarity matrix using the Jaccard indices [29]:

$$S_{c_i,c_j} = \begin{pmatrix} J(c_{i,1}, c_{j,1}) & \cdots & \\ \vdots & \ddots & \vdots \\ & \cdots & J(c_{i,n}, c_{j,m}) \end{pmatrix}. \quad (3)$$

The best match of cluster sets would yield few elements with value 1 and many with 0. Bad matches have many data points changing between clusters, leading to an array with many values $< 1$. Since we prefer to see dimension orders with many good matches, we subtract the matrix's mean value from all its elements $s_{x,y}$. We then square all results and sum them up into the grand sum to get a scalar similarity value between the sets of clusters:

$$similarity_{i,j} = \sum_{\substack{1 \le x \le n \\ 1 \le y \le m}} \left[ s_{x,y} - mean\left(S_{c_i,c_j}\right) \right]^2. \quad (4)$$

To obtain a cost function that fits the requirements of A* and our tree $T$, we invert the similarity. However, $sim(c_i, c_j)$ can be 0 when all matrix entries have the same value or we are comparing sets of size 1. Thus, we define the cost function as

$$horizontalCost_{i,j} = 1 + [similarity_{i,j} + 1]^{-1} \qquad \ge 1 \quad (5)$$

to avoid division by 0. Using our proposed metric helps the optimization algorithm in avoiding adjacent dimensions with no or only one cluster between them.

### 4.3 Cluster Ordering

We separate the optimization of horizontal dimension and vertical cluster order because CF-PCPs aim to primarily display the data flow between subspace clusters. The list of available clusters for display is controlled by the sequence of dimensions but not by their internal vertical arrangement. Therefore, the order of dimensions is our highest priority. Afterward, we still have flexibility in choosing the vertical arrangement along each data dimension. Our approach to optimizing the vertical sequence is similar to the dimension ordering: employ A* to find an optimal path from tree level 1 to a leaf at level $d$. The main difference lies in the way we generate the children of each node. When there are $m$ clusters to be arranged, we generate all permutations and add them to the parent node, which is itself a node among permutations of the previous clusters.

We adopted a metric based on the number of line crossings as a cost function—which is popular for ordering in PCPs [8]—and adapted it for the display of data flow from one cluster to another. The comparison of Figures 4b and 4c shows that lines within cluster pairs move closer to each other. This is caused by the axis scaling from Section 4.1 and creates the impression of bundles. Intertwined lines within such a bundle do not interfere with tracing the data flow, but crossing bundles are difficult to follow visually. Therefore, we define our metric to penalize these inter-bundle crossings.

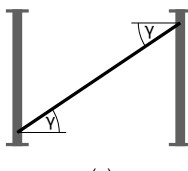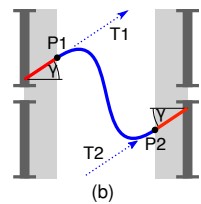

Figure 8: Composite line geometry in CF-PCPs. In regular parallel coordinates (a), lines have an angle $\gamma$, depending on the underlying data point's values in each display dimension. Lines in CF-PCPs (b) have segments with the same angle in a zone close to the axes (red lines on gray background). These are then connected by a cubic Hermite curve with tangents T1 and T2.

A bundle between clusters contains all shared points, e.g., $c_{i,1} \cap c_{j,1}$. We assume that each line has a certainty $l \in [0, 1]$ that describes its—possibly fuzzy—cluster membership. Fuzzy lines will not be as visible as certain ones and contribute less to overall clutter. See Section 5.3 for more details on how the certainty will affect line rendering. Our metric sums up the certainty within each bundle into $L$ and uses it as an adjusted line count. This new count is used to populate a matrix with values for all bundles between neighboring dimensions $i$ and $j$:

$$B_{i,j} = \begin{pmatrix} L(c_{i,1} \cap c_{j,1}) & \cdots & \\ \vdots & \ddots & \vdots \\ & \cdots & L(c_{i,n} \cap c_{j,m}) \end{pmatrix}. \quad (6)$$

Our approach calculates the total number of crossings using a method by Rit [43]: bundles only cross if they start lower and end higher (see Figure 7). The opposite case (from higher to lower) is just a different view on the intersection of the same lines. The actual calculation is done by multiplying each element $b_{x,y}$ of $B_{i,j}$ with the submatrix $_{x,y}B_{i,c}$ to its lower left and then getting the grand sum. The first column and last row do not have a submatrix to the lower left, so we do not need to calculate crossings for them. The resulting values are written into a matrix $I_{i,j}$ and all its elements $i_{x,y}$ are summed up. The final cost for placing one sequence of clusters next to another is then

$$verticalCost_{i,j} = 1 + \sum_{\substack{1 \le x \le n \\ 1 \le y \le m}} i_{x,y} \qquad \ge 1. \quad (7)$$

We start with a minimal vertical cost of 1 to ensure that the metric fits the requirements set by the level-based heuristic for A*. The width of the primary tree structure $T$ grows much faster than in Section 4.2. Therefore we recommend using the lazy A* algorithm for the optimization of up to 15 clusters per 2D subspace and reverting to a greedy approach for more complex problems. For example, sorting clusters by the mean value of contained data points approximates the vertical line layout from PCPs whilst retaining advantages of their cluster-flow counterparts.

## 5 RENDERING

At this point, CF-PCPs exist as a model that does not completely specify how to display lines and encode uncertainty. To be able to render actual visual output, we now address these aspects.

### 5.1 Curve Geometry

Traditional PCPs draw data points as straight lines between two neighboring dimensions. If we do the same in CF-PCPs, we get a relatively clean visualization without much clutter (Figure 9a). Unfortunately, the naive approach of axis duplication also changes the line slopes with respect to traditional PCPs. In fact, the slope depends no longer just on the data values but much more on the cluster

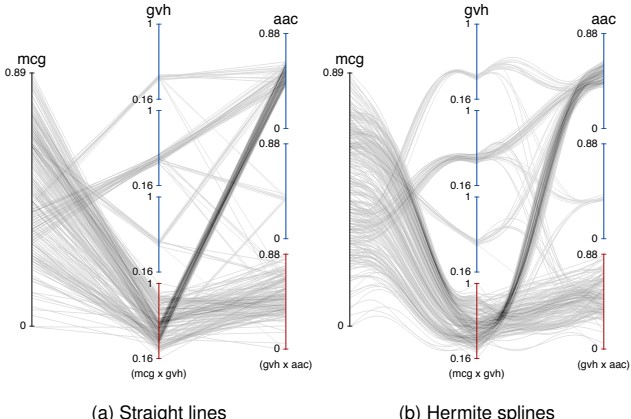

(a) Straight lines          (b) Hermite splines

Figure 9: Comparison of line geometry in CF-PCPs. Straight lines (a) make it hard to recognize correlations within clusters. Curved composite lines (b) preserve information on correlations by starting and ending with the same angles as in regular PCPs. Adjusting tangents used in the central Hermite splines separates lines that would cover and occlude each other completely.

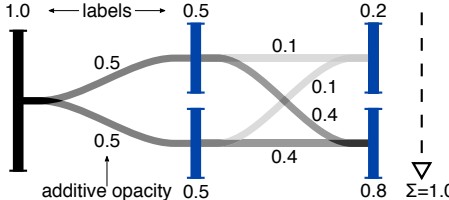

Figure 10: Fuzzy clustering provides a single data point with multiple soft labels in various clusters. Out technique draws lines between all possible pairs of these clusters. Their opacity is calculated by multiplying the point's labels in both connected clusters. This spreads each data point's constant total density of 1 over all possible connections.

membership. A related issue is that the angular resolution is reduced because lines become "compressed" (i.e., closer in angles) if they belong to the same cluster. Therefore, it becomes harder to recognize correlations. A recent eye-tracking study [37] showed that participants focus primarily on the area around the axes when reading PCPs. Another study with extreme bundling in PCPs [19, 20] showed that the perception of data characteristics and correlation is even possible when participants could not use the center parts between axes. These findings give us an opportunity to address the aforementioned problems: we replace straight lines by composite connections (see Figure 8). The key to our model is that we start and finish the connections with short straight line segments with the same slope as in original PCPs. This creates a zone in the very important focal area around the axes that retains the same information as in regular PCPs.

In the center region—in-between data axes—we connect the straight segments with a cubic Hermite curve. Choosing the tangents of the curve to be identical to the original slope guarantees a smooth transition between segments. The result of this approach is shown in Figure 9b. Instead of using the same factor to scale the Hermite curve's tangents $T1$ and $T2$, we vary it between data points. The relative index of each row in the source data set is added to a base factor on $T1$ and subtracted for $T2$. This spreads the composite lines along the direction they would have had in a classic PCP. This is very helpful when there are multiple row with the same values in both neighboring data dimensions: the wider the li-

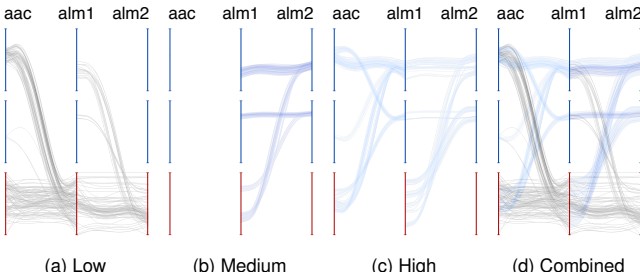

(a) Low          (b) Medium          (c) High          (d) Combined

Figure 11: Visualizing uncertainty in the clustering results. The label weights are binned into 3 ranges. Each bin is separately rendered to a density field and then mapped to color (a)–(c). Finally, the results are sorted by uncertainty and alpha blended (d).

nes spread, the more rows share the same values. The effect can be observed in the curves between the dimensions *Oil* and *Biomass* in Figure 15. In regular PCPs, this information would be lost in overplotting or would require a density rendering technique.

## 5.2 Density Rendering

CF-PCPs are designed to work with large data sets that can incur the problem of overplotting. We address this issue by adopting the established method of splatting [53] the lines and showing their density [2, 31]. This is achieved by splitting the rendering process into separate passes. The first pass uses additive blending to compute an intermediate density field. Since the densities typically cover a large dynamic range, we apply a nonlinear transformation before we render the final visualization. In our implementation, we apply a logarithmic mapping together with a logistic function to guarantee user-specified extrema within $[0, 1]$. The result is then multiplied with the line color's alpha value. The result is best demonstrated in Figure 1, where individual lines can be traced in areas of both high and low density. We discuss the choice of color in the following context of uncertainty visualization.

## 5.3 Uncertainty

Fuzzy clustering involves soft labeling of individual data points. In our implementation, we distinguished between no clusters at all (black axes in Figure 11), noise (red), and actual clusters (blue). This gives viewers additional information about the underlying algorithm's success in clustering the source data. Going further, more levels of certainty could be visualized as long as the axis colors remain well distinguishable.

Soft clustering can also select multiple labels for a single point, weakly assigning it to multiple clusters, instead of a single strong result. We treat these labels as probabilities and, by definition, the total probability that a point exists is 100 %, i.e., the sum of all assigned soft labels has to be 1. The clustering technique does not supply information on flow between sets of fuzzy clusters in separate subspaces. Therefore, we cannot infer whether a point moved between specific pairs of neighboring clusters. The probability that it belongs to one cluster or the other is our only information. We take this into account by rendering a single source data point as multiple lines to obtain a faithful visual representation. When a data row is labeled more than once in a 2D subspace, we draw a line between all neighboring clusters it belongs to (see Figure 10). We determine the opacity of each line by multiplying the label values of the clusters it connects. Our method corresponds to providing a field with probable paths for the movement of data points and results in an invariant total opacity of 1 for each point. This property is compatible with the rendering technique from Section 5.2 because it does not affect the total density.

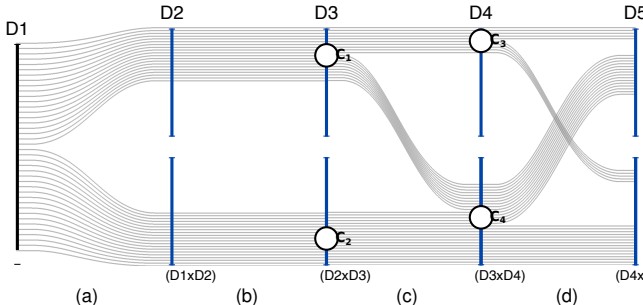

(a)  (b)  (c)  (d)

Figure 12: Inter-cluster patterns. Clustering in subspaces $D1{\times}D2$ and $D2{\times}D3$ gives the exact same result (a, b). Cluster $C_1$ splits into $C_3$ and $C_4$ in $D3{\times}D4$ (c). The displayed clusters in (d) have very low similarity with the results from the previous subspace.

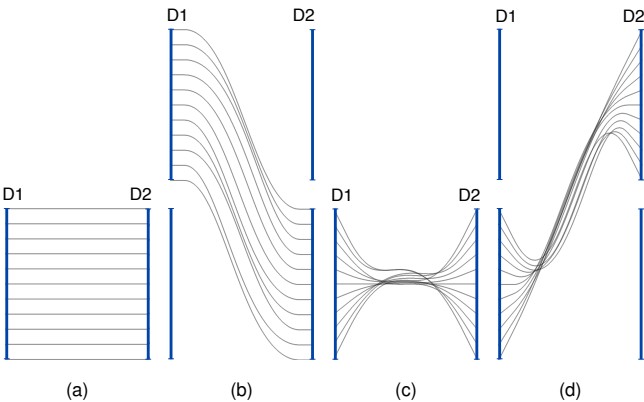

(a)  (b)  (c)  (d)

Figure 13: Intra-cluster patterns. When CF-PCP axes are at the same height (a and c), patterns from positive and negative correlations look very similar to their counterparts in regular PCPs. The line geometry discussed in Section 5.1 preserves these patterns close to the axes, even if they are at different heights (b and d).

Up to this point, our approach only renders the probability that a data point flows from one cluster to the other. Color is a strong visual variable with many distinguishable values, but we have not used it for line rendering, yet. Hence, CF-PCPs encode the numerical uncertainty from the clustering algorithm by mapping it to color. Each line between two axes is rendered according to its underlying data point's label in the target cluster (right axis for reading direction LTR). This means that the color of lines for the same data row may change from one side of an axis to the other, leaving the positional attributes as hints for tracing the line paths. We choose a sequential color scale (from light blue to black) to map the label weights in our examples (see Figure 11). The gradient from light blue to black only varies significantly in saturation and value, making it suitable even for viewers with most common color vision deficiencies.

CF-PCPs apply binning for good readability [38] because smooth color gradients can be problematic for reading exact values [4]. Another benefit of binning becomes evident when creating the density fields (see Section 5.2): additive blending would mix and distort the used colors. Furthermore, it would make multiple overlayed fuzzy lines look like a single certain one. To address these issues, our approach creates a separate density field for each bin in the uncertainty color map. As depicted in Figure 11, CF-PCPs convert them separately into regular images with the RGB values from the color map, while the alpha channel encodes the field's density. In the final render pass, these individual fields are sorted by fuzziness and alpha-blended with the *over* operator [42].

## 6  VISUAL PATTERNS

As discussed earlier, CF-PCPs show streams of 2D clusters across the subspaces of a data set while still displaying the original correlations between dimensions. As such, our visualization aims at providing two levels of granularity: overview and detail. The overview level shows similar 2D subspaces, clusters, and inter-cluster patterns, i.e., how data flows between clusters across different subspaces. The detail level shows intra-cluster patterns, i.e., what is the correlation within pairs of clusters from neighboring dimensions.

**Ordering:** Our horizontal ordering, coupled with our A* approach, implies strong similarities of clusters in neighboring dimensions because we minimize the total dissimilarity according to our metrics. More specifically, this means that the two subspaces between three neighboring dimensions contain similar clusters.

**Clusters:** The duplication of axes allows for easy identification of the number of cluster within each 2D subspace. In turn, this helps us identify data classes across different pairs of dimensions. Coupled soft labeling, this makes it easy to recognize well-separated classes (with a high level of certainty) over fuzzy ones.

**Inter-Cluster Patterns:** We created Figure 12 as an example with five dimensions that always have two clusters in their subspa-

ces. In Figures 12a and 12b the flow of data between clusters in $D1{\times}D2$ and $D2{\times}D3$ shows highly similar subspaces: each cluster on the left is associated with a unique cluster on the right. This one-to-one relationships denotes high or total similarity between subspaces. The situation is different in Figure 12c, where the CF-PCPs shows that cluster $C_1$ from $D2{\times}D3$ splits into $C_3$ and $C_4$ in $D3{\times}D4$. It would also be correct to say that $C_4$ splits into $C_1$ and $C_2$, depending on the point of view. Finally, in Figure 12d, we see two very dissimilar subspaces. Here, both clusters $C_3$ and $C_4$ split and mix in $D4{\times}D5$.

More generally, if each cluster in the subspace of two dimensions is uniquely associated with every other cluster of a neighboring subspace (i.e., the similarity matrix in Equation (3) contains only ones and zeros), it means that they behave similarly. If, in turn, all clusters are linked with each other (i.e., the similarity matrix contains many values $<1$ and $>0$), this shows a completely dissimilar clustering behavior between the three dimensions that span the two subspaces. Additionally, an axis having different incoming clusters is equal to a merge or split, depending on the point of view. Overall, in CF-PCPs, the number of splits or merges between clusters conveys the connectedness of their two subspaces.

**Intra-Cluster Patterns:** Atop showing the high-level information on flow of data between clusters across subspaces, our visualization also maintains patterns known from traditional parallel coordinates. Since the duplication retains the entire value range of an axis, our cluster-flow layout shows the distribution of each cluster's data points in the original dimensions. As demonstrated by the clusters labeled $D3$ in Figure 1, we can see where the values are positioned in regards to the whole range: larger values are in the upper cluster, smaller ones in the lower. This approach also visualizes whether the data points move closer or further apart, e.g., the bundle from $D3$ spread out in $D4$.

The examples in Figure 13 illustrate how CF-PCPs show data correlations. If two neighboring clusters are aligned (a and c) then positive and negative correlations look very similar to the classic PCP approach. In the case of non-aligned clusters (b and d), our line geometry from Section 5 ensures that the slopes near the axes remain identical to the ones in regular PCPs. Hence, our approach preserves the most important areas of PCPs [37] and adds additional information on clusters and data flow.

Overall, the visual patterns of our approach can be seen in different places of the visualization: inter-cluster patterns are located in the entire space between two display dimensions, whereas the vicinity of duplicated axes reveals intra-cluster patterns.

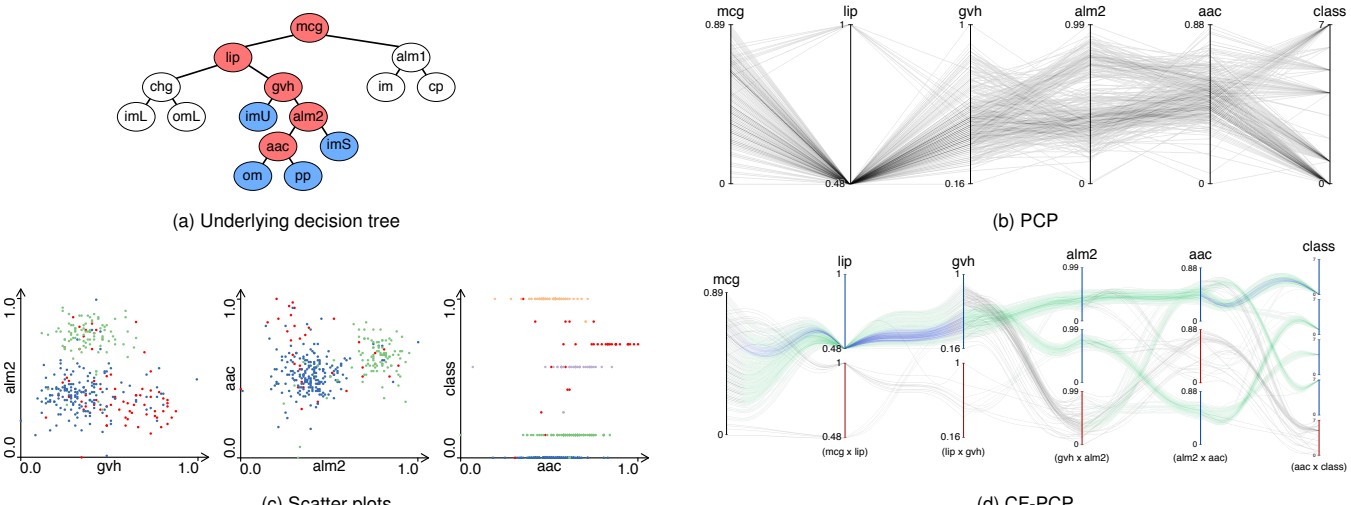

(a) Underlying decision tree

(b) PCP

(c) Scatter plots

(d) CF-PCP

Figure 14: Classification of E. coli bacteria according to a tree (a) where nodes correspond to metrics and leafs to inferred classes [24]. All red metrics are necessary to distinguish the blue classes. These are shown as dimensions in the regular PCP (b). The scatter plots show color-coded fuzzy subspace clusters between the last three metrics (c). Our CF-PCP (d) technique combines aspects of both previous visualizations. Its line colors encode certainty: 0 ▇▇▇ 100 %.

## 7  EXAMPLES

To demonstrate the applicability of CF-PCPs, we provide examples for typical real-world data sets and compare our results to previous approaches.

### 7.1  Escherichia Coli

Horton and Nakai used machine learning to automatically predict localization sites of proteins [24]. Their results included a decision tree that uses multiple measured metrics for Escherichia coli bacteria in order to arrive at a predicted classification. Figure 14a shows this tree and highlights five red nodes that we selected for visualization. They correspond to the metrics that define four classifications (blue). We start our analysis with a regular PCP and density rendering (Figure 14b). It shows that the dimension *lip* is only binary. There seem to be negative correlations between *alm2* and its neighbors, but it is difficult to map the resulting class to a specific influence of each previous dimension. The CF-PCP in Figure 14d uses 2D subspace clustering and shows additional information. The first impression is of mostly green color, which shows low certainty. This is due to a low degree of separation between the clusters. The second impression is of flow between clusters. The observable flow from the CF-PCP matches the tree structure from Figure 14a. Using Fuzzy DBSCAN with dimensions *mcg*, *lip*, and *gvh* yields a single cluster and noise each time. The classification tree confirms that they are not sufficient to infer any specific classes. Subspace *gvh*×*alm2* shows the first split into two actual clusters, just as the tree also arrives at its first class *imU*. Class *imS* is inferred next and the CF-PCP also contains two clusters in *alm2*×*aac*. Combining the information from *aac*×*class* reveals four final clusters, which matches the the classification count in the highlighted subtree.

The scatter plots with color-coded clusters do not show this data flow between dimensions because the selected colors are not linked between each plot. Even the display of the fuzziness poses a problem: some points belong to multiple clusters. Plotting them multiple time on the same position only retains the color of their last label.

### 7.2  NetPerf

To help understand the characteristics of CF-PCPs, we also compare our technique with two clustering-oriented PCP approaches

in Figure 2, using the NetPerf data set [49]: Illustrative Parallel Coordinates (IPC) [36] and an edge bundling layout (EBL) for interactive parallel coordinates [39]. In contrast to our proposed pairwise fuzzy method, the clustering in IPCs is calculated and rendered for all dimensions at once (see Figure 2c). Conversely, the EBL method clusters the data separately in each single dimension (see Figure 2a). Both approaches use color to encode clusters consistently over every dimension. Our selected sample visualizations show four dimensions and three clusters in the NetPerf data set. To increase comparability, we limit ourselves to the four dimensions used in the previously published visualizations of the same data. Dimension ordering is disabled, while the vertical axis order minimizes our metric on line crossings from Section 4.3.

In Figure 2c, Illustrative Parallel Coordinates bundle PCP polylines based on the cluster they belong to. This reduces inter-cluster overdraw and emphasizes visual cluster separation. Line distortion is applied at the expense of clarity of pairwise axis correlation, because their direction no longer corresponds to the original angle in regular PCPs. Clustering in IPCs works across all visible dimensions and shows overlap between clusters, for instance, in the signal strength dimension. In contrast, EBL [39] reduces clutter in PCPs by bundling edges within pairs of one-dimensional clusters that are computed with a *k*-means algorithm. Similarly to our approach, this method highlights subspace clusters—albeit one-dimensional ones—and allows viewers to follow the flow of bundles between dimensions. As such, the method avoids overdraw of clusters on axes. However, the one-dimensional approach simplifies the visualization to the detriment of details when it comes to diverging clusters. For example, edges in the top cluster of the throughput axis split into three different clusters in signal strength, but without further interaction, users cannot infer the distribution of data points that belong to the highest cluster in the signal strength dimension.

Similarly to both alternative approaches, our cluster-flow layout shows three main clusters per axis pair (see Figure 2b). However, our visualization goes beyond the alternative techniques and is able to show overlap between clusters while allowing the viewer to trace complex cluster patterns. More specifically, our plot shows that low framerates are always associated with low throughput. Just as with logical consequence, the opposite is *not* true. Similarly to IPCs (Figure 2c), the CF-PCP shows that the main cluster of high

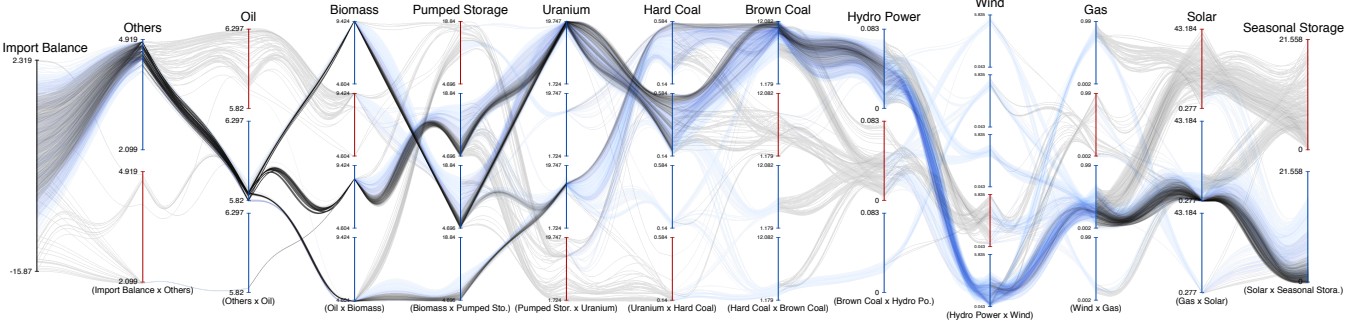

Figure 15: Visualization of electric power production by primary energy sources in Germany. Clusters in this CF-PCP are vertically ordered by their mean value. Line colors encode certainty from fuzzy clustering: 0 ▆▆▆ 100 %

throughput is associated with three clusters of high to low values of signal strength. Contrary to the broad bands in EBLs, fine details in CF-PCPs allow the viewer to see that the highest values of signal strength are always associated with the highest values of throughput. The large cluster at the top of the throughput dimension in the EBL of Figure 2a does not support this conclusion.

A commonality with the EBL is the top cluster between signal strength and throughput, which forks into two different clusters when combining the data on signal strength and bandwidth: one with high, the other with low values. In the CF-PCP, we can quickly determine that there are many data points with high framerate, despite having only medium to low throughput and minimal signal strength and bandwidth. Color coding fuzziness, we are also able to see that there are uncertain assignments, e.g., the third cluster between framerate and throughput is completely fuzzy and many of its data points are more likely to be noise or belong to the second cluster. Overall, our implementation avoids the inconvenience of overlap from *n*-dimensional clustering and allows for detailed analysis by tracing individual lines and clusters across dimensions.

### 7.3 Energy Production

Lastly, we visualize a larger data set of electricity production by primary energy source [13, 44]. It contains 1750 data rows and lists a timestamp, 12 sources of primary energy, as well as international imports and exports as dimensions we can use in the CF-PCP. While the restriction to subspaces generally reduces the number of clusters, this is not the case with the uniformly distributet data in the time dimension: it yields 12 clusters. We removed it to get a cleaner and less cluttered plot for further analysis. We provide further visualizations of the original and reduced data set as supplemental material.

Clustering between pairs of dimensions and separating them vertically can help with visual detection of dependencies in the data. From Figure 15, we can extract information between hard and brown coal. On the one hand, low energy production from the latter only occurs when hard coal also has lower values. On the other hand, power production on the left-hand side varies greatly, while brown coal is often at a high output level. Therefore, it seems that between these two, brown coal is burned more constantly while hard coal is preferred for adjustments to changing levels in power production from unsteady sources. Even with density rendering, the high degree of overplotting would create an almost constant background between both axes and thus cannot facilitate these observations.

### 8 Conclusion and Future Work

We have presented a technique for cluster-centric visualization of high-dimensional data using parallel coordinates. The main idea of our technique is to deliberately duplicate axes for each cluster to show data flow between 2D subspaces. At the same time, this also creates an opportunity to display fuzzy clusters with parallel coordinates. We have described an algorithm to compute the cluster-flow layout, i.e., for ordering dimensions and axes. While the automatic optimization is algorithmicly complex, manual interaction with the axes and dimensions is always a viable alternative for data exploration. We analyzed visual correspondence of data patterns and discussed the applicability of our technique using multiple examples. Our layout is an improvement over the traditional approach when clusters are not linearly separable over a single dimension or all dimensions together and the number of clusters in subspaces is small: in this case, we bundle lines over separate axis clones instead of plotting many overlapping lines on a single large axis.

In future work, we want to thoroughly evaluate user performance in controlled studies. In a first step, CF-PCPs should be compared against regular PCPs and scatter plot matrices (SPLOM) [10, 18] separately. A second step would compare them against a combination of both, for example, in coordinated views. It might also be interesting to investigate whether the vertical position of the largest cluster influences user performance.

Extensions of our work might look into alternative optimization targets for the layout, such as aesthetic aspects or faithfulness. Parallel coordinates and scatter plots have their strengths and weaknesses. We would like to integrate scatter plots into our layout, e.g., beneath or between duplicated axes. A progressive clustering and cluster-flow layout would also be of interest for the analysis of dynamic data with live updates. We used CF-PCPs with a reading direction, where each display dimension is used for clustering with its neighbor. Our method could be extended to use both neighbors. Another change could be to cluster all displayed dimensions with a common reference dimension. This would be similar to selecting a row or column from a scatter plot matrix and visualizing it with parallel coordinates. Considering the example with E. coli bacteria, a hierarchical approach would be very beneficial for small numbers of dimensions. Here, we would enforce a reading direction and progressively create subclusters that mimic the classification tree from Horton and Nakai's work [24]. Finally, we would like to expand our approach to the field of interactive model peeling in the context of regression and machine learning.

### Acknowledgments

Funded by the Deutsche Forschungsgemeinschaft (DFG, German Research Foundation) – project ID 251654672 – TRR 161 (projects A01 and B01). We would like to thank Prof. Dr. Bruno Burger (Fraunhofer Institute for Solar Energy Systems ISE) for his cooperation in the analysis of energy production data.

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
