# OpenReview forum: "Cluster-Flow Parallel Coordinates: Tracing Clusters Across Subspaces"
_graphicsinterface.org/Graphics_Interface/2020/Conference — GI 2020_

### Official Review · AnonReviewer3 · 2020-02-11
**Paper is interesting; thorough studies are necessary.**

**Rating:** 6
**Confidence:** 3

**Review:**

This paper is a novel visualization of parallel coordinate plots by condensing lines/flows when there exists a cluster or trend in a subspace of the data appearing in 2D scatter plots. To make this visualization possible, some extra axes are duplicated when a cluster presents in the data.

The paper is very well-written, visualizations are good looking and the method seems mathematically robust. However, I have serious problems with the usefulness of the approach in practice.

In this novel representation, as shown in Fig 2, values of data points are omitted. One of the good properties of parallel coordinate plots is that following the chart, we can understand the value of each data point and also its scale. However, this is not possible in the proposed visualization.

In parallel coordinate plots, a line takes the data from one dimension to another. In this new approach, the line is replaced by a Hermite curve trying to respect the angles that the line makes with the vertical axes. However, I find it a bit confusing specially because Hermite curves sometimes produce peaks in the curve that do not help the clarity of visualization.
It also took me some time to understand the role of the extra stacked axes in these plots. The same is true for the fuzzy color coding. I cannot make sure that these replacements can actually increase the readability of the plots.

Considering the above issues, I think that it is necessary to perform thorough user studies and quantitative studies that why these design decisions help the readability of  parallel coordinate plots. Although performing such studies is left for the future work by the authors, I think the usefulness of this new visualization is under question without having such studies. The examples that are provided in the paper along with their interpretations and comparisons are interesting but I am not sure that an external user or observer can understand the data as nicely and interpret them similarly.

Since the paper is well written and it has some merits, I am okay with publishing the paper if other reviewers find it a good fit. I believe the paper is on the border-line.



Minor issue:

The paper would be more clear if the method was discussed using a simple numerical data instead of only visual data (Fig 5).

---

### Official Review · AnonReviewer1 · 2020-02-12
**No evaluation although the work is interesting**

**Rating:** 6
**Confidence:** 3

**Review:**

The authors present a cluster-centric visualization of high-dimensional data using parallel coordinates. It duplicates axes for each data to show data flow between subspaces. The authors also present a description of a layout algorithm. Uses cases of visualization are provided. The paper claims that the technique is an improvement over traditional approaches but without a user study this is difficult to know. I would have rated the paper higher if there was at least a preliminary evaluation. So this is a borderline paper.

On the positive side, the presentation of work is of high quality.

---

### Official Review · AnonReviewer2 · 2020-02-12

**Rating:** 6
**Confidence:** 3

**Review:**

The paper proposes a new visualization scheme that combines the properties of scatterplots and parallel coordinates plots (PCPs): the Cluster-Flow Parallel Coordinates Plot (CF-PCP).  The visualization represents clusters of data points in multivariate data by duplicating axes from the canonical PCP visualization to represent 2D subspaces of the multivariate data.  This approach preserves the readability of correlational patterns from the original PCP while making cluster assignments more obvious than alternatives relying on edge bundling and on just the use of line color.

The implementation of the proposed visualization requires tackling several interesting aspects including a scheme to connect lines between duplicated axes by drawing Hermite spline segments that preserve the line slopes at the axes and a layout optimization based on an A* algorithm to compute the shortest path ordering of duplicated axes.  The results are demonstrated on several example datasets and contrasted against visualizations using traditional PCP and scatterplots.

This is a nice paper that I believe proposes and novel and useful visualization scheme.  However, there is one key weakness which prevents me from being more positive with respect to acceptance: an evaluation of the proposed visualization in practical use through a user study is absent.  The benefits of the visualization are only demonstrated through qualitative results.  The paper would have been significantly stronger if the expected benefits were measured in a practical scenario.

---

### Meta-Review · Area_Chair1 · 2020-02-13

**Recommendation:** Accept
**Confidence:** 3

**Metareview:**

All the reviewers agree that this is a good piece of work but all have concerns that the usefulness of the approach in practice has not been proven due to the absence of evaluation. All reviewers also agree that the quality of the presentation of the work is one of its strong merits.

---

### Decision · Program_Chairs · 2020-02-18

Accept